# Efficient Co-Valorization of Phosphogypsum and Red Mud for Synthesis of Alkali-Activated Materials

**DOI:** 10.3390/ma16093541

**Published:** 2023-05-05

**Authors:** Qingsong Liu, Xiangci Xue, Zengqing Sun, Xiaoxian Huang, Min Gan, Zhiyun Ji, Xuling Chen, Xiaohui Fan

**Affiliations:** School of Minerals Processing & Bioengineering, Central South University, Changsha 410083, China

**Keywords:** phosphogypsum, red mud, alkali-activated materials, mechanical property, reaction mechanism

## Abstract

Phosphogypsum and red mud are bulk industrial solid wastes that trigger local environmental problems. In the present investigation, an efficient valorization process was developed through which phosphogypsum and red mud can be transformed into a precursor for the synthesis of high-strength, alkali-activated materials with a seawater-bearing sodium silicate solution as the alkaline activator. The effects of the activator modulus and liquid-to-solid ratio on the strength evolution of the synthesized AAMs as well as the microstructure and chemistry of the reaction products were investigated. The results showed that mineral reconstruction between PG and RM took place during calcination at 950 °C, forming ye’elimite, anhydrite and gehlenite, which then took part in the alkali-activation process and generated thenardite and C-A-S-H gel. The mechanical properties of the synthesized AAMs, ranging from 12.9 MPa to 40.6 MPa, were determined with the activator modulus and liquid-to-solid ratio. Results from the present investigation contributed to the facile and efficient valorization of phosphogypsum and red mud into cementitious construction materials.

## 1. Introduction

Phosphogypsum (PG) is the by-product of fertilizer production from phosphate ore. Depending strongly on the composition and quality of the phosphate ore, PG is typically powdery with little or no plasticity and predominantly composed of calcium sulphate [1,2]. For the production of every ton of phosphoric acid, normally 4–6 tons of PG are generated [3]. Approximately 200 million metric tons of PG are annually generated with a utilization rate of 10–15%. The remaining 85–90% is stockpiled in landfills. Global stockpiles of phosphogypsum have exceeded 6 billion tons [4], which not only occupy considerable land areas but also cause serious environmental problems [5]. The utilization of PG is becoming a focused problem and attracting more and more general concern.

Extensive practices have been made in using PG as a cement-setting regulator, replacing natural gypsum [6]. Singh ground cement clinker with PG and found a significant increase in the initial and final setting times [7]. The delayed setting was assigned to the impurities, such as F^−^, P_2_O_5_, organic matter, etc., that form a protective coating on the surface of cement particles. To diminish PG’s negative effect on setting time, pretreatment methods, such as washing, wet sieving, neutralization, calcination, etc., have been developed [1,4,8], and the setting time of PG and water-washed PG-blended cement was tested. Compared with the control sample using natural gypsum, PG resulted in considerable enhancement of the setting time while the water-washed counterpart caused a marginal increment. Moreover, the incorporation of calcined PG at 200–800 °C contributed to a decrease in the delayed setting, and the declination increased with the increase in calcination temperature, though all samples possessed a longer setting time than cement containing natural gypsum [9]. A similar phenomenon was found in terms of the mechanical properties of PG-incorporated cement concrete. In [7], mixing PG with cement clinker led to the corresponding reduction in 1-, 3-, 7- and 28-day compressive strength by 37.5%, 42.86%, 37.5% and 18.48%, respectively. Meanwhile, strength enhancement was detected when pretreated PG was used. An increase in the 28-day flexural and compressive strengths of 9.55% and 7.1%, respectively, was reported by Shen et al. [8], in which washed PG was incorporated.

Meanwhile, PG has been extensively reported to worsen the workability of obtained cement [4,10,11], and the declination increases with an increase in PG content. This phenomenon was related to the density difference among PG, cement and aggregates [4] and has been regarded as a defect of utilizing PG in cement.

Alkali activated materials (AAMs), a class of clinker-free cementitious materials derived by the reaction of aluminosilicates with alkaline medias, have received increasing attention from the academic and industrial fields due to their excellent mechanical properties and significantly reduced CO_2_ emissions [12]. The aluminosilicate sources for AAM production range from industrial byproducts/waste to calcined clays, including metallurgical slag, fly ash and metakaolin. Recently, new precursors, such as mineral processing tailings, sludge incineration products and agriculture wastes, have also been developed for AAM production [13]. Extensive investigations have demonstrated that AAMs exhibit the superior engineering characteristics of high early and final strengths, fire and thermal resistance, low permeability, adhesiveness to different surfaces, etc. [14,15]. Moreover, the estimated CO_2_ emission of AAMs can be up to 90% less than OPC of comparable strength grade [15,16]. Benefiting from these advantages, AAMs are considered one of the most alternative binders for the sustainable development of the construction industry [12,13,14,17,18]. The application of AAMs in storehouses, masonry blocks, drainage systems, pavement, etc., has been made since the 1970s. Significant results have been achieved by Australia on the commercialization of AAMs. In 2013, a four-story building containing suspended AAM concrete floors was constructed and put into service. One year later, the Brisbane West Wellcamp Airport (BWWA), constructed using approximately 40,000 m^3^ of AAM concrete, was opened for business [19,20].

Rashad investigated the possibility of partially replacing fly ash with calcined PG (CPG) for the synthesis of AAM [21]. Replacing FA with 5% and 10% CPG was demonstrated to be beneficial for strength development, while a replacement of 15% resulted in strength reduction. Moreover, after being exposed to thermal treatment at 400–1000 °C, the AAMs containing 5% and 10% CPG exhibited 2.1 times and 1.7 times greater residual strength than control paste, respectively, suggesting that the CPG helped to improve the fire resistance of AAMs. Similar results were reported by Vaiciukynienc et al. [22] when adding PG into alkali-activated GGBS. In [23], PG was blended with blast furnace slag and used as a precursor for AAM synthesis. Complete dissolution of PG was detected, demonstrating its participation in the alkali-activation process. Moreover, incorporating PG contributed to enhance the compressive strength of the obtained AAMs, which can be raised from the accelerated formation of the amorphous binding gel phase and the change in gel chemistry. Though positive results as well as confidence for recycling PG with an alkali-activation process have been achieved by the above individual investigations, the incorporation amount is limited. Efforts are required to boost the large-scale utilization of PG.

The Bayer process is the main method used in China for bauxite treatment for alumina production, and red mud (RM) is the associated byproduct [24]. According to statistics, the annual production of RM is approximately 150 million tons all over the world [25]. RM is normally composed of Fe_2_O_3_, Al_2_O_3_, SiO_2_, CaO, TiO_2_ and some trace elements. Several hydrometallurgy and pyrometallurgy methods have been developed to recycle the iron, aluminum, titanium and scandium [26,27,28,29]. Other potential utilization, such as sewage treatment, cement and concrete production, catalyst, etc., has also been developed and well-reviewed in [30]. Meanwhile, the utilization rate of RM is far from satisfactory.

In the present investigation, an innovative and efficient co-valorization of PG with RM is developed through which both industrial wastes can be converted into a precursor of high reactivity for the synthesis of AAMs. The physicochemical properties of PG, RM and the prepared precursor were studied. The influence of the activator modulus and liquid-to-solid ratio on the mechanical strength of the synthesized AAMs was investigated. The microstructure and chemistry of the reaction products were well characterized to gain insight into the reaction mechanism. The results obtained in the present investigation contribute to the fast valorization of PG and RM and the sustainable development of cementitious construction materials.

## 2. Materials and Methods

### 2.1. Materials

The PG used in the present investigation was provided by a phosphate fertilizer company in Yichang, China. The RM was obtained from an aluminum company in Binzhou, China. X-ray fluorescence (XRF, EAGLE III, Green Bay, WI, USA) and X-ray diffraction (XRD, X’Pert Pro Diffractometer, Almelo, The Netherlands) were applied to determine the chemical and mineralogical compositions of both materials with the results shown in Table 1 and Figure 1, respectively. The PG is predominantly composed of CaO and SO_3_, which are in the form of gypsum. The composition of RM is complicated, in which Fe_2_O_3_, Al_2_O_3_, SiO_2_ and Na_2_O are the main components. As can be inferred from Figure 1, the Fe_2_O_3_ and Al_2_O_3_ in RM are mainly present as hematite (Fe_2_O_3_) and goethite (FeO(OH)), gibbsite (Al_2_O_3_·3H_2_O) and boehmite (Al_2_O_3_·H_2_O), respectively. In addition, phases, including quartz (SiO_2_), sodalite (Na_4_(Al_3_Si_3_O_12_)Cl), rutile (TiO_2_), anatase (TiO_2_) and phlogopite (KMg_3_[AlSi_3_O_10_](OH)_2_), were also detected.

The particle size distribution of PG and RM, measured using Malvern Instruments (Mastersizer 2000, Worcestershire, UK), is shown in Figure 2. The PG is composed of particles ranging from 0.3 μm to 160 μm with the D10, D50 and D90 being 4.932 μm, 28.422 μm and 72.264 μm, respectively. In contrast, the particle size of RM locates in a broader range of 0.1–200 μm, and the corresponding D10, D50 and D90 are 0.766 μm, 2.819 μm and 42.244 μm, respectively, indicating that the RM is slightly finer than the PG.

The NETZSCH instrument (STA-449F3, Netzsch, Germany) was applied to characterize the thermogravimetry (TG) and differential scanning calorimetry (DSC) properties of PG and RM. As shown in Figure 3, the mass loss of PG is 9.7% and can be classified into three stages, i.e., 25–110 °C, 110–200 °C and 200–1000 °C. The mass loss up to 110 °C is 0.62%, which can be assigned to the evaporation of free water. The predominant mass loss occurred at the 110–200 °C stage, accounting for 74.34% of the total mass loss. This stage is related to the decomposition of gypsum into anhydrous calcium sulphate and the loss of the hydrated water, which is also accompanied by a sharp endothermic peak at 159 °C. Afterwards, the mass loss was very slight. It should also be mentioned that the TG curve slope increased at > 800 °C. This might be attributed to the evaporation of inorganic salts such as NaF.

In comparison, continuous mass loss from 25 °C to 900 °C can be observed in the TG curve of RM, which includes the evaporation of free water and decomposition of gibbsite and gothite [31]. All of these were further confirmed by the DSC results. The endothermic peak at 280 °C supports the decomposition of gibbsite and the formation of boehmite. Cryptocrystalline or transition of alumina might also take place at this temperature range [32,33]. The decomposition of goethite into hematite as well as the transformation of boehmite into alumina can be reflected as the shoulder at 323 °C and the broad peak at 400–550 °C, respectively. Another broad endothermic peak can be clearly observed at 800–1000 °C accompanied by a very slight mass loss. This might be attributed to the phase transformations and/or sintering of RM.

### 2.2. AAM Synthesis

Prior to the synthesis of the AAMs, PG and RM were activated to prepare a precursor of high reactivity. The activation is derived from our previous work [31,34]. In brief, weighted PG and RM were thoroughly mixed via ball milling; the mixture was then shaped into pellets of 10–15 mm in dimension followed by calcination at 950 °C in a Muffle furnace for 30 min. The precursor for AAM synthesis was obtained by cooling down the calcined pellets to room temperature naturally and grinding to pass through a 0.074 mm sieve. The mass ratio of PG/RM is 1.67, which is on the basis of favoring active phase reconstruction; details are referred to [31,34]. The calcination temperature and time were demonstrated to be efficient for the formation of mineral phases of high reactivity.

The AAMs were synthesized by mixing the obtained precursor with an alkaline activator. The activator used was a sodium silicate solution containing 29.5 wt.% SiO_2_, 11.7 wt.% Na_2_O and 58.8 wt.% H_2_O, which was obtained by dissolving silica fume and NaOH pellets into seawater. The seawater was collected from Beibu Gulf, China. The solution modulus (SiO_2_/Na_2_O molar ratio) was adjusted to 1.5, 2.0 and 2.5 by further adding NaOH pellets (analytical grade). The influence of the liquid-to-solid ratio on the properties of the AAMs was also investigated, which was 0.35, 0.45 and 0.55. In total, AAMs with nine proportions were prepared. The sample ID is titled as ‘AAM-activator modulus-liquid/solid ratio’; for example, AAM-1.5–0.45 represents a specimen synthesized using an activator with a modulus of 1.5 and a liquid–solid ratio of 0.45. The mixing of the solid precursor with the alkaline activator was conducted in a planet mixer for 2 min. Standard sand, 3 times the precursor mass, was added into the paste and mixed for another 3 min. Afterwards, the homogeneous mortar was casted into molds of 40 × 40 × 40 mm^3^, vibrated 2 min to remove air bubbles, covered with a glass plate and stored in a curing box of 20 °C and approximately 100% RH. Samples were demolded after 24 h and stored at room temperature. The compressive strength was measured at the ages of 3, 7 and 28 d.

### 2.3. XRD Measurement and Data Analysis

The mineral compositions of PG, RM, the prepared precursor and synthesized AAMs were determined using powder XRD (X’Pert Pro Diffractometer, PANalytical, Netherlands). For the measurement of the synthesized AAMs, pastes were prepared following the same preparation and curing protocol described in Section 2.2 without adding standard sand. Before the XRD measurement, pastes were crushed and ground according to the RILEM TC-238 method [35].

The theta–theta geometry was applied for the XRD measurement with the 2θ ranging from 5° to 60°. The tube operating condition was 40 kV and 40 mA to generate CuKα X-ray. Data analysis was conducted using the software HighScore Plus from PANalytical; more details are referred to [36,37].

### 2.4. SEM–EDS Measurement

The microstructures of the prepared precursor and synthesized AAMs were determined using a TESCAN MIRA3 SEM. The measurement of the AAM was conducted using the paste sample with the purpose to better characterize the morphology and composition of the reaction products. Samples were coated with gold in advance of the measurement.

## 3. Results and Discussion

### 3.1. Mineral Composition of the Prepared Precursor

Figure 4 depicts the XRD pattern of the prepared precursor using PG and RM as the raw materials. The reflections of gypsum in PG together with the peaks of boehmite, gibbsite, sodalite and phlogopite in RM almost disappeared. New reflections of ye’elimite, anhydrite and gehlenite are detected. Ye’elimite is one of the key components of calcium sulfoaluminate (CSA) cement. The synthesis of ye’elimite has been well investigated in [38,39]. Normally, the calcination of calcium aluminate phases (CA, CA_2_) with CaSO_4_ at 1000–1300 °C is required to obtain ye’elimite of high purity. In the present investigation, reflections of ye’elimite were determined under calcination at 950 °C, which is similar to the results reported in [31]. This might be caused by the formation of a low-melting mixture system due to the presence of fluxes, such as SiO_2_, MgO, Na_2_O, etc. The formation of ye’elimite is temperature dependent; increasing calcination temperature could obviously accelerate its formation [31]. The anhydrite in the precursor should be the result of the dehydration of gypsum in PG during calcination, which can extensively take place at 110–200 °C (Figure 3). Formation of gehlenite has been frequently reported during the co-thermal treatment of RM with gypsum and other materials [19,40]. No other hydraulic phases were detected in the present investigation because of the relatively low calcination temperature and the low CaO content in the raw materials. This is also favored by the fact that hematite is found as the main iron-bearing phase.

### 3.2. Morphology of the Prepared Precursor

The morphology of the prepared precursor was characterized using SEM and is shown in Figure 5. The compositions of selected areas were determined with EDS and are listed in Table 2 for which the back-scattered electron was used to clearly display the contrast. The ground precursor grains are of irregular structure with a size up to 10 μm. It can be inferred from Table 2 that the local chemical compositions of the precursor powders vary significantly. Even a single grain can be divided into areas with different composition characteristics. Figure 5ii clearly shows areas of grey, medium light and light property, demonstrating that different mineral phases are embedded together. The grey areas are mainly composed of Ca and S with 10.3% Fe and a slight content of Si, Al and Na. With the enhancement of light, the ratio of Fe increased considerably to 25.47% (point F) and 49.85% (point G). Combined with the XRD results, point G might be assigned to hematite, and a similar conclusion can also be made for points I and M.

In comparison, the composition of point F is more complicated, and the minerology can hardly be decided. This is the situation for most of the characterized grains, including points A, B, C, D, H, J and L. In point E, the grain exhibits very smooth and uniform morphology. The particle is predominantly composed of Ca, S and O with the molar ratio of Ca/S being 0.98. Thus, this section might be attributed to anhydrite.

### 3.3. Strength of the Synthesized AAMs

Figure 6 depicts the compressive strength of the synthesized AAM mortars. The compressive strength increased from 4.1–16.7 MPa to 12.9–40.6 MPa as the curing time extended from 3 d to 28 d. Both the activator modulus and liquid–solid ratio have a considerable effect on the development of compressive strength. As can be observed from Figure 6, the 28-day compressive strength of specimens AAM-1.5-0.4, AAM-2.0-0.4 and AAM-2.5-0.4 was 40.6 MPa, 25.1 MPa and 14.7 MPa, respectively. In contrast, the compressive strength of the AAMs produced with the same activator decreased with an increase in the liquid–solid ratio. The changes can be related to the activator alkalinity and water involved. As the solution modulus increased from 1.5 to 2.5, the Na_2_O content decreased from 18.3% to 12.1%. An activator with a lower modulus indicates a higher content of Na_2_O and alkalinity, which involuntary accelerates the dissolution of the solid precursor and formation of the reaction products [36]. Moreover, the corresponding water content in the activator with a modulus of 1.5, 2.0 and 2.5 is 55.2%, 57.2% and 58.6%, respectively. Relatively high water content can contribute to the diffusion of dissolved ions at the very early stage of the alkali-activation process but will lead to increased porosity and have a negative effect on the mechanical properties of the obtained AAMs [36,37]. Thus, the highest compressive strength in the present investigation was achieved by specimen AAM-1.5-0.4, which is consistent with our previous works [17,31,37,41]. Much higher compressive strength was realized by calcining the sulfate-rich solids with RM at higher temperature [31]. This was not adopted in the present investigation partially because the strength meets a broad range of engineering application requirements and facile thermal treatment possesses the benefits of reduced energy consumption and CO_2_ emission.

### 3.4. XRD of the Synthesized AAM

Figure 7 depicts the XRD pattern of the synthesized AAM paste after 28 d curing. Compared with the precursor, changes that include the disappearance of ye’elimite and gehlenite as well as the significant intensity decrease in anhydrite can be clearly observed, demonstrating that these phases took part in the alkali-activation process. Ye’elimite is reported to be a key reactive component in sulfoaluminate cement, which decreased rapidly after contact with water [40]. Dissolution of gehlenite and anhydrite during the alkali-activation process is also reported in [31,36]. In contrast to the dissolution of these components, new reflections related to thenardite were detected. The alkaline activator provided a considerable amount of free Na^+^ to the reaction system, which can easily combine with the SO_4_^2−^ released from the dissolution of ye’elimite, anhydrite and other sulfate bearing components, precipitating as thenardite during the hardening of the paste. The hematite and quartz might be more stemmed from the precursor, which is relatively inert during the alkali-activation process [17,36].

In addition to the crystallized phases, hump centering at approximately 22–38° 2θ can be observed. This can be attributed to the calcium aluminosilicate hydrate gel (C-A-S-H), which is the main reaction product and strength-giving phase of AAMs with a high calcium content [14,17,36]. The initial high content of CaO in the mixture favors the formation of C-A-S-H gel. Though frequently regarded as aluminum substituted C-S-H gel, C-A-S-H is different from C-S-H both in chemical composition and microstructure. C-A-S-H is reported to possess a denser structure than C-S-H, contributing to better mechanical characteristics of the corresponding system [40].

### 3.5. Microstructure of the Synthesized AAM

The microstructure of the synthesized AAM paste is illustrated in Figure 8a. Dense structural characteristics can be observed, which are responsible for the excellent mechanical behavior mentioned above. In general, fine precursor grains with individual morphology were well interlinked by a dense gel-like matrix. Some cracks can also be observed, which might have originated from the sample preparation process. The thin specimen used for the SEM measurement was obtained by mechanical crushing of the cured AAM paste and then drying at 40 °C in a vacuum environment. Throughout the sample preparation, mechanical stress and water evaporation can induce cracks.

To verify the composition of the synthesized AAM, EDS mapping was conducted with the results shown in Figure 8b–i and Table 3. The obtained AAM is mainly composed of O, Na, Ca, Si, Fe, S and Al but with different distribution characteristics. Areas with enriched Fe and Al can be inferred from the relative brightness of the EDS mapping image. The Fe-rich particle might be assigned to hematite, which remains inert during the alkali-activation process. Several of these particles can be observed being incorporated randomly into the AAM paste. This is consistent with the XRD results in which hematite is one of the main products in the synthesized AAM paste. The Al-rich area can be related to Al(OH)_3_, which can be stemmed from the reaction of ye’elimite and other Al-bearing phases. Formation of Al(OH)_3_ was also detected during the hydration of the calcium sulfoaluminate-ferrite clinker in Hertel et al. [40]. This reaction product is of low content, which cannot be effectively detected via XRD. In contrast, the distribution of the other elements is more even. Significant overlapping of the elements Na and S can be inferred from a close check of Figure 8g,h, which can be related to thenardite (Na_2_SO_4_), a key reaction product of the synthesized AAM (detected with XRD in Figure 7). Similarly, the considerable overlapping area of Ca with Si and Al exhibits gel-like characteristics, which can be attributed to C-A-S-H. This confirms the assertation in Section 3.4, but a cross-check with other advanced characterization techniques is required to obtain quantitative analysis results and promote an understanding of the gel chemistry. In general, the gel matrix exhibits dense characteristics and is well-connected with hematite and thenardite, contributing to the excellent mechanical property of the synthesized AAM.

## 4. Summary and Conclusions

In the present investigation, phosphogypsum and red mud were synergistically converted into alkali-activated material. The influence of the activator modulus and liquid-to-solid ratio on the mechanical behaviors of the synthesized AAMs was investigated. Attention was also paid to the micro-properties of the synthesized AAMs. Based on the discussion above, the following conclusions can be drawn:During the calcination at 950 °C, mineral reconstruction between PG and RM took place. Gypsum, boehmite, gibbsite, sodalite and phlogopite in the source materials were converted into ye’elimite, anhydrite and gehlenite.The mechanical properties of the synthesized AAMs were significantly influenced by both the activator modulus and liquid–solid ratio. The 28-day compressive strength of the synthesized AAMs was up to 40.6 MPa.The main reaction products of the synthesized AAMs were thenardite and C-A-S-H gel. The synthesized AAMs possessed a dense microstructure in which the reaction products and unreacted particles were well interlinked by the gel matrix, contributing to the excellent mechanical performance of the synthesized AAMs.

## Figures and Tables

**Figure 1 materials-16-03541-f001:**
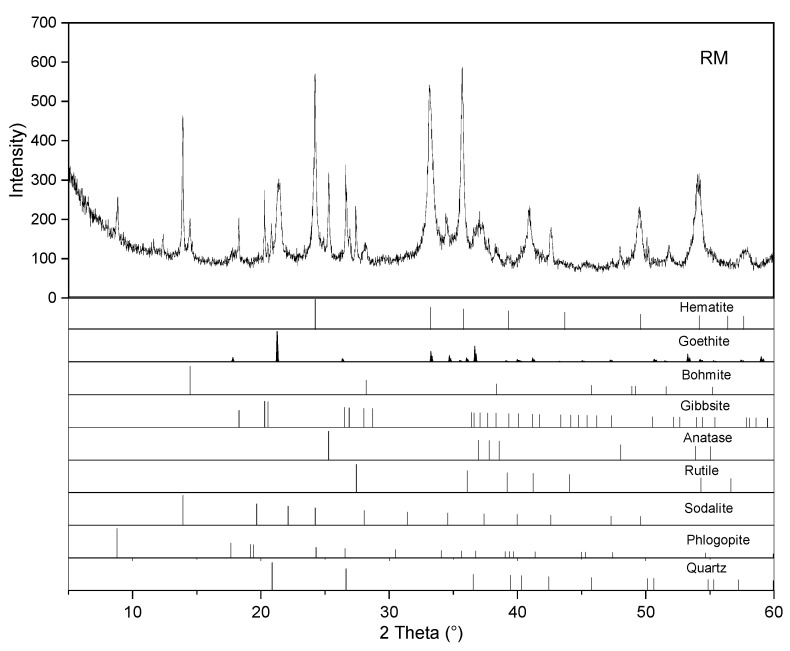
XRD patterns of PG and RM.

**Figure 2 materials-16-03541-f002:**
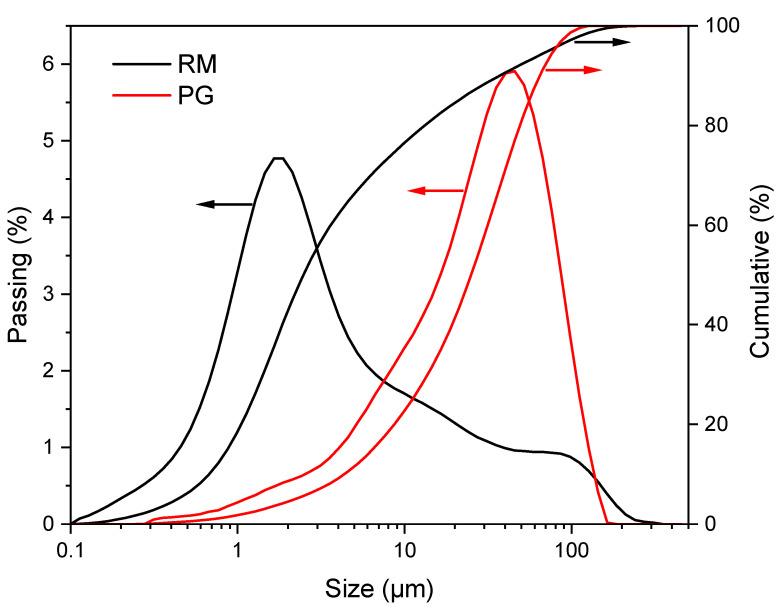
Particle size distribution of PG and RM.

**Figure 3 materials-16-03541-f003:**
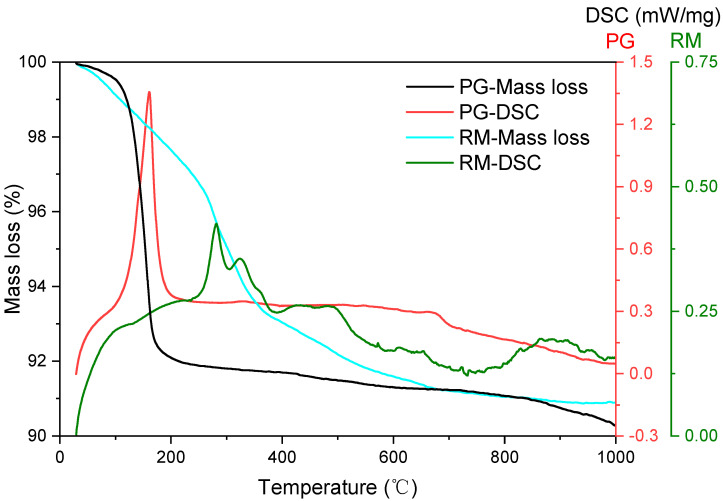
TG–DSC curves of PG and RM.

**Figure 4 materials-16-03541-f004:**
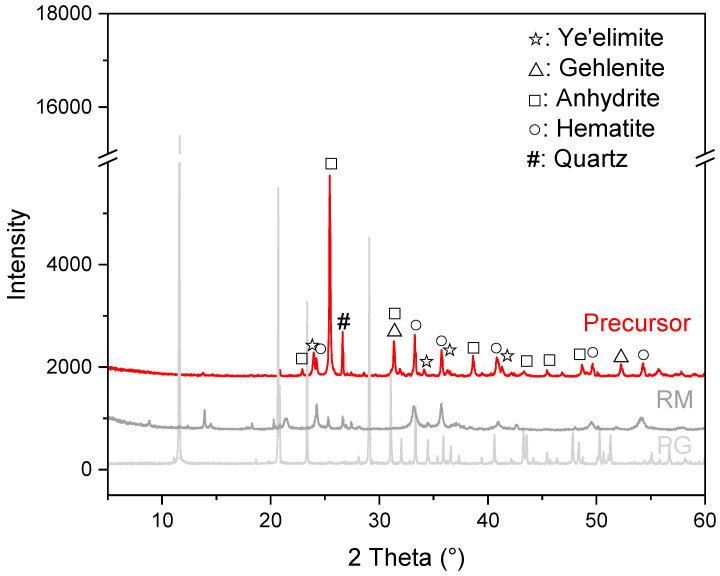
XRD pattern of the prepared precursor.

**Figure 5 materials-16-03541-f005:**
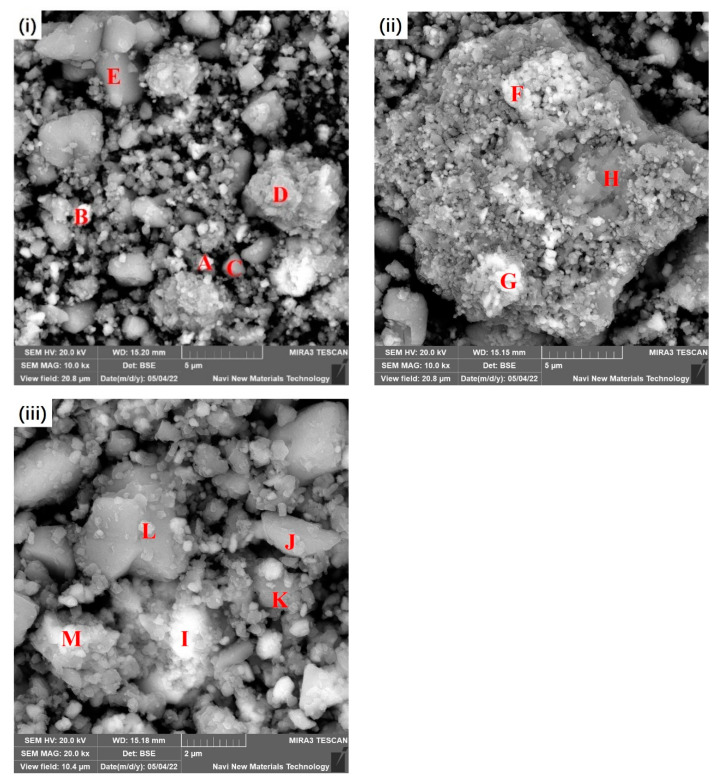
SEM of the prepared precursor.

**Figure 6 materials-16-03541-f006:**
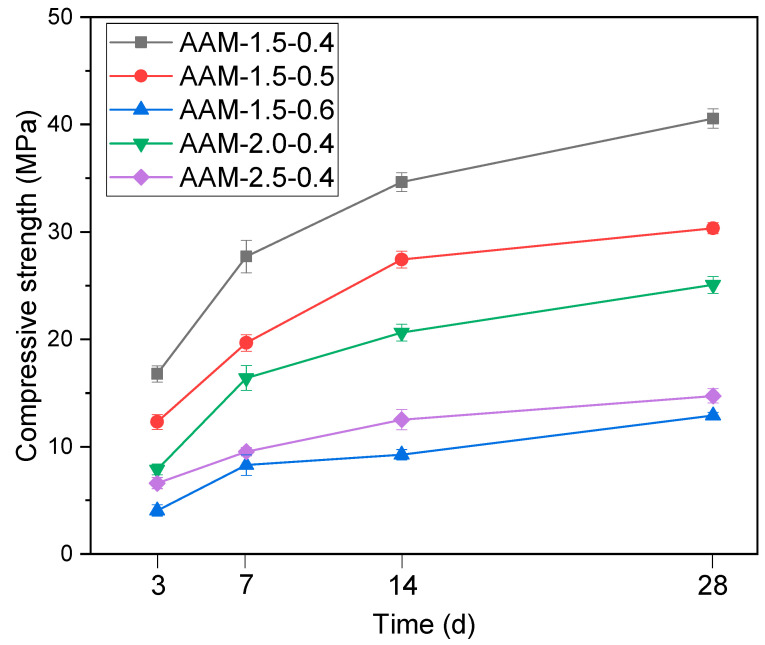
Compressive strength of the synthesized AAM mortars.

**Figure 7 materials-16-03541-f007:**
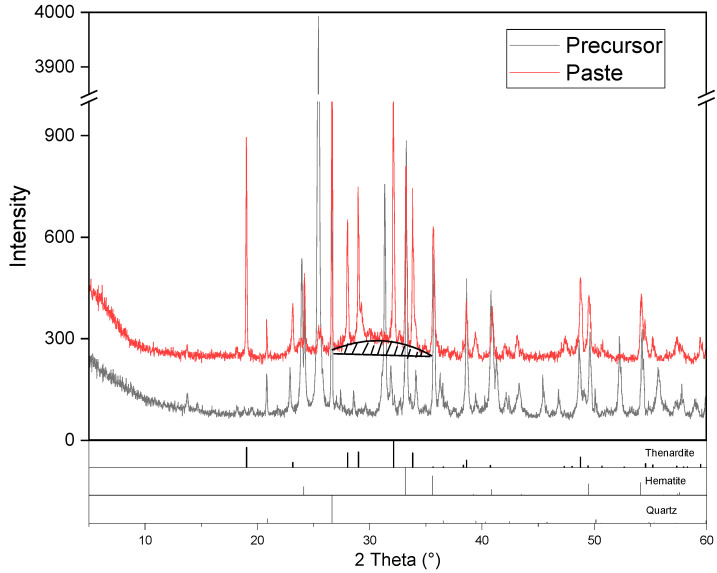
XRD pattern of the synthesized AAM.

**Figure 8 materials-16-03541-f008:**
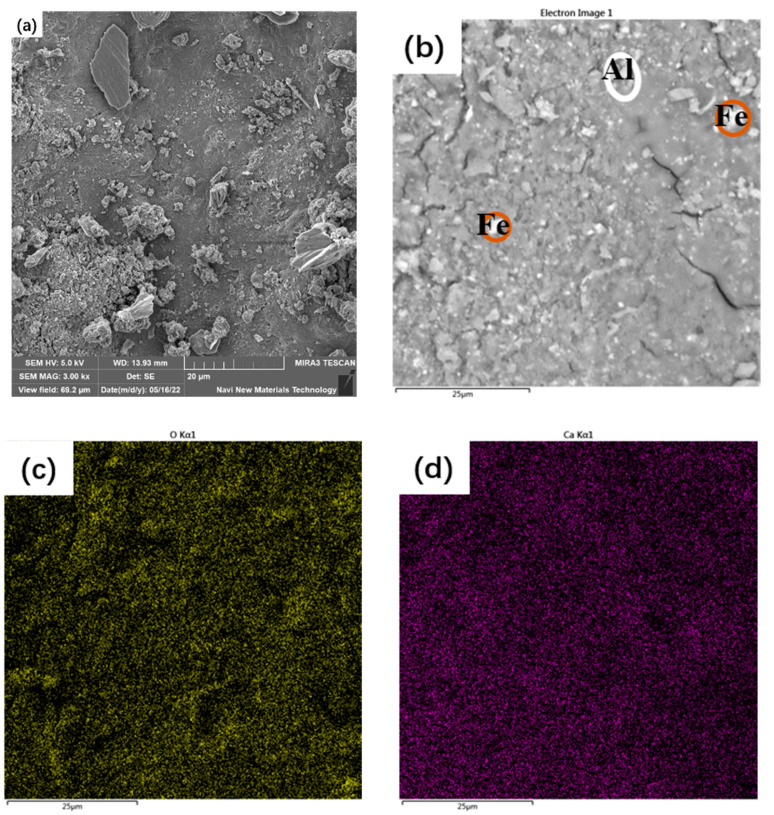
SEM and EDS mapping of the synthesized AAM.

**Table 1 materials-16-03541-t001:** Major chemical composition of PG and RM given in % by weight.

	Al_2_O_3_	SiO_2_	CaO	Fe_2_O_3_	MgO	TiO_2_	Na_2_O	K_2_O	P_2_O_5_	SO_3_
PG	1.04	2.22	35.46	0.66	0.62	0.05	0.02	0.13	1.52	21.51
RM	22.1	15.8	1.7	37.1	0.1	5.4	11.6	0.2	0.03	0.63

**Table 2 materials-16-03541-t002:** Chemical compositions of selected areas in the obtained precursor shown in Figure 5 (wt.%).

Areas	O	Na	Mg	Al	Si	S	Ca	Ti	Fe
A	53.85	3.05	0.74	3.77	1.64	1.61	2.53	1.42	31.34
B	43.91	2.41	0.19	2.90	1.53	5.78	6.35	0.34	36.52
C	49.45	2.31	0.08	3.05	1.80	14.95	20.89	0.48	6.92
D	47.55	4.31	0.14	6.79	4.44	9.37	13.28	0.82	13.06
E	53.41	0.79	0.03	1.01	0.45	19.23	23.59	0.12	1.35
F	46.05	4.55	0.40	8.89	3.89	3.74	5.97	0.97	25.47
G	37.69	2.04	0.63	2.68	1.33	1.38	2.78	1.57	49.85
H	29.07	1.64	0.06	4.47	3.62	19.30	29.60	1.64	10.30
I	26.90	2.46	1.05	5.12	2.33	3.40	4.48	2.16	52.02
J	56.66	2.03	0.21	3.33	1.70	13.07	16.52	0.56	5.87
K	44.73	2.47	0.22	6.12	3.51	10.56	16.20	1.18	14.89
L	55.53	1.72	0.17	2.98	1.72	14.14	17.29	0.41	6.01
M	37.41	2.82	1.02	4.82	1.78	2.39	3.17	1.87	44.68

**Table 3 materials-16-03541-t003:** Chemical compositions of the EDS mapping areas shown in Figure 8 (wt.%).

Element	O	Na	Al	Si	S	Ca	Ti	Fe
Content	41.19	16.95	3.07	11.16	7.37	11.45	0.71	7.88

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
