# Peer review of "Efficient Co-Valorization of Phosphogypsum and Red Mud for Synthesis of Alkali-Activated Materials"

_materials, 2023, doi:10.3390/ma16093541_

Round 1

Reviewer 1 Report

I have gone through the manuscript entitled "Efficient Co-valorization of Phosphogypsum and Red Mud for Synthesis of Alkali-activated Materials". The work is well presented and methodology provided is sufficient to replicate the work. However, some modifications is needed for better understanding.

1. The word "in this work" is repeated many time, which is understood. It is suggested to the authors kindly use the present investigation and only where it is necessary.

2. Line no: 240 the underline need to be removed.

3. Figure 5 the embedded text is not clear. The clear image should be provided.

4. In line 392 "in" is repeated so need to be corrected.

5. Line no: 402-409 should be written as "recommendation or Future work"

Author Response

1. The word "in this work" is repeated many time, which is understood. It is suggested to the authors kindly use the present investigation and only where it is necessary.

Answer: We thank the reviewer for the critical comment. This has been revised as commented.

2. Line no: 240 the underline need to be removed.

Answer: We are sorry for the mistake. The underline has been removed.

3. Figure 5 the embedded text is not clear. The clear image should be provided.

Answer: Thanks for the comment. Clear images have been provided in the revised manuscript.

4. In line 392 "in" is repeated so need to be corrected.

Answer: This has been revised following the comment.

5. Line no: 402-409 should be written as "recommendation or Future work".

Answer: Line 402-409 has been removed to keep the manuscript clear and concise.

Reviewer 2 Report

The research subject is interesting. However, there are some drawbacks  and some drawbacks, as stated in popup notes inside the text, which I recommend to include the AAM control sample.  Or,  give an appropriate respond why the authors did not do so.

Author Response

The research subject is interesting. However, there are some drawbacks and some drawbacks, as stated in popup notes inside the text, which I recommend to include the AAM control sample.  Or,  give an appropriate respond why the authors did not do so.

Answer: We thank the reviewer for the critical comment. The control sample is presented in another paper published in Construction and Building Materials. To avoid any copyright problem, the control sample is not included in present manuscript, but the findings related to the control sample has been cited in present investigation.

Reviewer 3 Report

Manuscript ID: materials-2340089

Title: Efficient Co-valorization of Phosphogypsum and Red Mud for Synthesis of Alkali-activated Materials

Authors: Qingsong Liu et al.

Line 40-44. The authors use two references to reference [7]. As the sentences follow each other, perhaps one link should be removed.

Line 84-88. Can the authors provide information on newer constructions using AAM concrete?

Line 108-109. “for bauxite refining” – it is an incorrect term. Please, rewrite this sentence as follows: Bayer process is the main method used in China for bauxite treatment for alumina production

Line 108-117. Authors are encouraged to use more relevant links from 2021 to 2023.

Table 1. The sum of oxides is not 100 wt.%, why?

Line 140-142. Authors should add the chemical formulas of the minerals in brackets.

Figure 1. The authors should enlarge the figures, they are too small.

Figure 3. Authors should separate the TG/DSC to each sample. It is difficult to understand the information on one figure.

Figure 6. Usually in articles about construction materials, this information is presented as a bar chart. Where there are 4 columns for each point on the X-axis.

Table 2-3. Authors add information on chemical composition of precursor and synthesized AAM using SEM/EDS. Why didn't the authors use the XRF method?

References are not in Materials style. Please write information for authors: https://www.mdpi.com/journal/materials/instructions

Author Response

Line 40-44. The authors use two references to reference [7]. As the sentences follow each other, perhaps one link should be removed.

Answer: We are sorry for the mistake. The second one has been removed.

Line 84-88. Can the authors provide information on newer constructions using AAM concrete?

Answer: There are new constructions using AAMs, such as retaining wall surrounding a bridge, residential housing slab, etc., in Australia, India and China. But compared with the Brisbane West Wellcamp Airport, theses buildings are of smaller volume and lower popularity. These new constructions are cataloged by https://www.geopolymer.org/, which were not listed in present investigation.

Line 108-109. “for bauxite refining” – it is an incorrect term. Please, rewrite this sentence as follows: Bayer process is the main method used in China for bauxite treatment for alumina production …

Answer: We thank the reviewer very much for the comment. This sentence has been revised following the comment.

Line 108-117. Authors are encouraged to use more relevant links from 2021 to 2023.

Answer: Newer and more relevant links from 2021 to 2023 are used in the revised manuscript.

Table 1. The sum of oxides is not 100 wt.%, why?

Answer: As titles, Table 1 only listed major components, trace elements are not shown here, which do not possess influence on the thermal activation and alkali activation process.

Line 140-142. Authors should add the chemical formulas of the minerals in brackets.

Answer: We thank the reviewer for the comment. Chemical formulas of the minerals are added.

Figure 1. The authors should enlarge the figures, they are too small.

Answer: According to the comment, figures have been enlarged.

Figure 3. Authors should separate the TG/DSC to each sample. It is difficult to understand the information on one figure.

Answer: Thank for the comment. We tried to separate the TG/DSC to each sample, but this would cause the comparison and data analysis more difficult. We presented the TG and DSC of RM and PG indifferent colors and well labeled to make this figure easy to understand and compare.

Figure 6. Usually in articles about construction materials, this information is presented as a bar chart. Where there are 4 columns for each point on the X-axis.

Answer: We appreciate the reviewer’s time and effort. The strength information can be displayed in several methods. We believe that readers can easily obtain strength evolution of AAMs in the way we present. This is also our way for publications in journals such as Cement and Concrete Research, Cement and Concrete Composites, Construction and Building Materials.

Table 2-3. Authors add information on chemical composition of precursor and synthesized AAM using SEM/EDS. Why didn't the authors use the XRF method?

Answer: XRF provides the total chemical composition of precursor and synthesized AAM, while Table 2-3 shows the detailed composition information of selected points/areas under SEM to help the analysis of local chemical and mineral compositions.

Round 2

Reviewer 3 Report

Article can be accepted in the present form.